# Temperature-mediated shifts in feeding behaviour and metabolism in an omnivorous rock pool prawn

Gustavo M. Martins[1,2], Christopher D. G. Harley [ID][3,4*], Ana Neto[2†], Francisco Arenas [ID][5]

**1** CIBIO, Research Centre in Biodiversity and Genetic Resources, InBio Associate Laboratory, University of the Azores, Ponta Delgada, Portugal, **2** Faculty of Sciences and Technology, University of the Azores, Ponta Delgada, Portugal, **3** Department of Zoology, University of British Columbia, Vancouver, British Columbia, Canada, **4** Institute for the Oceans and Fisheries, University of British Columbia, Vancouver, British Columbia, Canada, **5** Benthic Ecology and Environmental Solutions Team, Interdisciplinary Centre of Marine and Environmental Research (CIIMAR/CIMAR), University of Porto, Matosinhos, Portugal

† Deceased.
* harley@zoology.ubc.ca

## Abstract

Temperature is a key factor influencing metabolic processes and ecological interactions in ectothermic organisms, and can determine pathways of energy flow in food webs. In this study, we tested the hypothesis that omnivore diets shift away from carnivory and towards herbivory at warmer temperatures. Specifically, we investigated the effects of temperature on the feeding behaviour and metabolic activity of the omnivorous rock pool prawn *Palaemon elegans* Rathke, 1837, across three temperature regimes (15°C, 20°C, and 25°C) commonly encountered in intertidal rockpools of the Azores. *Palaemon* individuals were offered either algae (*Ulva muscoides*), amphipods (*Hyale perieri*), or a choice of both to assess feeding preferences. Our results demonstrate that temperature significantly influenced both the consumption of *Ulva* and *Hyale*, as well as prawn oxygen consumption. At lower temperatures (15°C), *Palaemon* consumed less *Ulva* when given a choice, while at higher temperatures (25°C), the consumption of *Hyale* decreased. Oxygen consumption increased significantly with temperature, indicating heightened metabolic activity in warmer conditions. These findings suggest that as temperatures rise, *Palaemon* may shift its feeding preferences towards herbivory, with potential implications for intertidal food webs in a warming climate. This study highlights the importance of temperature in shaping trophic interactions and food web structure, and underscores the need for further investigation into the long-term impacts of climate change on coastal ecosystems.

## 1. Introduction

Temperature plays a crucial role in biological systems by influencing a wide range of functions, from enzymatic kinetics to large-scale patterns of species biogeographic

**Data availability statement:** All data for this project are available through the Open Science Framework at https://doi.org/10.17605/OSF.IO/QRKAT.

**Funding:** Funding was provided through FCT (Fundação para a Ciência e a Tecnologia) under the projects UID/BIA/00329/2013 and UIDB/50027/2020 to AN. GMM was supported by FCT grant SFRH/BDP/108114/2015. FA received support through the Programa Operacional Regional do Norte (NORTE-01-0145-FEDER-031893). CDGH was supported by Natural Sciences and Engineering Research Council (Canada) Discovery Grants (RGPIN-2016-05441 and RGPIN-2022-04683).

**Competing interests:** The authors have declared that no competing interests exist.

distribution [1–4]. It is considered one of the most important ecological factors, influencing everything from body size to species abundance to ecosystem-level carbon flux [5–11]. Ongoing changes in the Earth's climate and associated fluctuations in temperature have intensified our interest in understanding the effect of temperature on all aspects of biological systems.

One topic that has garnered renewed interest is the role of temperature in determining the sources and rates of resource aquisition, and how energy flows through a foodweb based on the trophic relationships among species [12,13]. Ecological theory predicts a number of temperature dependences in food webs, including an increase in per capita consumption rates and a reduction in carrying capacity with increasing temperature [14]. The strength of trophic interactions can be strongly temperature-dependent [15,16], and because not all trophic levels are equally susceptible to temperature changes [17], temperature can have wide-ranging effects on community structure [18–20]. These effects create ecological patterns in space and time. For instance, the trophic level of copepod communities shifts across seasonal thermal gradients as herbivory increases relative to carnivory in the summer [21]. Across spatial thermal gradients, herbivorous fish are more abundant and have stronger impacts on algal resources in warm-water tropical regions compared to cooler, higher latitudes [22–24].

The observation that herbivory tends to predominate in warmer conditions, both within and among species of consumers, is related to the relationships between temperature and organismal metabolic demands and digestive effeciency. In ectotherms, temperature can variably influence different components of metabolic function, such as feeding, gut passage and assimilation rates [25,26], as well as respiration and growth [27,28]. Crucially for marine herbivores, the activity of cellulase, which aids in the efficient digestion of cellulose in plant material, is reduced below 20°C [29]. This constraint, coupled with the higher prevalence of herbivorous fish and greater feeding rates in warmer waters [13,22–24,30,31], suggests that marine omnivorous ectotherms may switch towards increased herbivory at higher temperatures, particularly when a threshold in the range of 15–20 °C is crossed.

In recent years, a small but growing body of research has experimentally explored the idea that temperature can affect feeding behaviour. While ectothermic herbivores and carnivores generally ingest more food with rising temperature (e.g., see review by [32]), omnivorous ectotherms may respond in two different ways: quantitatively by consuming more food and qualitatively by altering their degree of herbivory or carnivory through diet shifts. Shifts in diet in omnivorous ectotherms have been implied by seasonal variations in trophic level (using stable isotope analysis, e.g., [21]), by recording changes in the amount of different prey items ingested with rising temperature [e.g., 30,33], or by using choice *vs.* no-choice experiments, but usually using dead animal prey [13,34], which may be unrealistic in a natural setting.

In this study, we investigated the effect of temperature on respiration rates and algal vs. animal prey preferences of the omnivorous rock pool prawn *Palaemon elegans* Rathke, 1837, using live prey. The prawn *P. elegans* is a widespread species in coastal waters of Europe where it feeds on a diverse array of items, including

detritus, algal material (such as *Cladophora*, Ectocarpales and *Ulva*) and animals (such as gammarids, harpaticoids, chironomids and ostracods) [35,36]. Here, we conducted our trials at three seawater temperatures (15, 20, and 25 °C) that span the range of seasonal variation these animals experience in nature and bracket the putative herbivory threshold near 20 °C. After quantifying rates of oxygen consumption, the prawns were fed algae (no choice), amphipods (no choice) or a mixture of algae and amphipods (choice), and their consumption rates were compared. We predicted an increase in metabolic rate and a shift in dietary preferences away from animal prey and towards algal prey at higher temperatures.

## 2. Materials and methods

### 2.1. Palaemon elegans

The prawn *Palaemon elegans* Rathke, 1837, commonly known as the rockpool prawn, is a species of shrimp in the family Palaemonidae. It is found in the Northeast Atlantic, Mediterranean, Black Sea, and, more recently, the Baltic Sea [37,38]. In some areas of its geographical distribution, it is the most abundant Palaemonidae prawn [39] and is believed to play an important role in the trophic web by foraging on various food sources, and in turn, being consumed by predators. This omnivorous prawn feeds on a wide array of plants, animal taxa and detritus [35,36]. In the rockpools of the Azores, it can reach high populations densities [40,41].

### 2.2. Laboratory experiment

Individuals of *Palaemon elegans* were collected with a handnet from intertidal rockpools in Lagoa (37° 44' N, 25° 35' W), kept in water and promptly transported to the laboratory inside coolers. There, they were sorted by size, with small individuals (< 5 mm carapace length) being discarded. Larger individuals (those with a carapace length > 5 mm) were retained (n ≈ 100 individuals) and haphazardly divided into three groups, visually ensuring that each group had a similar range of sizes. Since there was no hypothesis regarding how temperature-dependent changes in diet should vary with the sex of prawns, these were sorted among treatments disregarding their sex. Any potential effect of sex was thus randomly distributed among treatments, so that it was unlikely a confounding factor. Prawns were acclimated for two weeks prior to the start of the experiment in temperature-controlled chambers under a 12:12 day-night cycle. Only those *P. elegans* individuals that had not moulted during the previous week were selected for the main experiment. Within the temperature chambers, each individual prawn was kept in an individual 0.5 L tank filled with continuously aerated seawater and fed *ad libitum* with commercial shrimp food (Hikari). The tanks were cleaned, and water (previously conditioned to the corresponding treatment's temperature) was replaced every other day.

The day before the experiment, *Ulva muscoides* and the amphipod *Hyale perieri* were collected from the intertidal zone. Both the alga *U. muscoides* (hereafter *Ulva*) and *H. perieri* (hereafter *Hyale*) are abundant species in the Azorean rocky intertidal and were readily consumed by *P. elegans* (as confirmed in a pilot study prior to the experiment).

The experiment had three treatments where *Ulva* (no choice), *Hylae* (no choice) or a mixture of *Ulva* and *Hyale* (with choice) were offered to prawns, with each treatment crossed with three temperature conditions (15, 20 and 25°C). There were 8 replicates per treatment combination, resulting in a total of 72 experimental units. The temperature range tested is commonly experienced by organisms in rockpools in the Azores and roughly corresponds to the average range of seawater surface temperatures in the region (Instituto Hidrográfico) (Fig 1). *Ulva* (mean ± SE, 0.102 ± 0.001 g wet mass) and *Hyale* (n = 10) were added to the experimental tanks according to the treatments during the light-off phase of the cycle. *Ulva* mass was measured using a precision scale after excess water was removed by shaking the alga and gently pressing it between two pieces of soft paper for 10 seconds. The quantities of *Ulva* and *Hyale* offered to the prawns was based on [36] and were pre-tested to ensure that neither food source was limiting over the experimental period. The consumption of *Ulva* and *Hyale* was estimated after 24 hours as the difference in mass (*Ulva*) and number (*Hyale*) remaining in the experimental tanks. No moulting occurred during the 24-hour experimental period. After the experiment, the prawns were

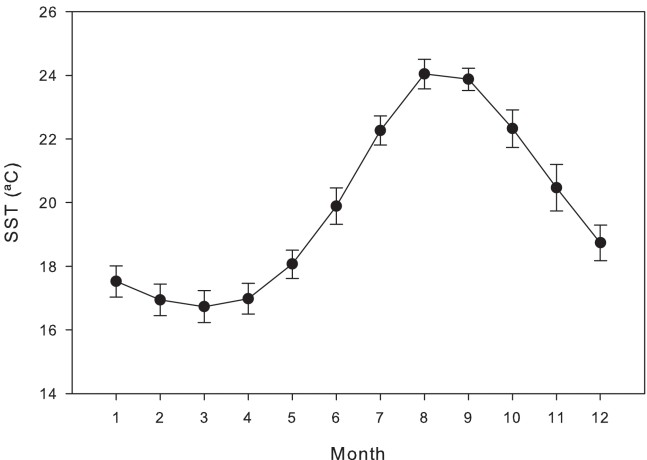

**Fig 1. Monthly mean (± SD, n = 19) sea surface temperature (1996-2014) in the Azores.**

euthanised by lowering temperature to 4°C until insensible and then rapidly cutting through the centreline of the head and tail. These were then dried (60°C) in an oven to constant mass, and weighed.

Additionally, oxygen consumption was measured in all tanks containing *Palaemon* one day prior to the experiment, as a proxy to estimate the effect of temperature on prawn metabolic activity. Oxygen consumption was assessed using an oxygen probe (Hanna HI 2004-02), both before and after an approximately 90-minute period during which water aeration was halted. Experimental tanks were kept open during incubations, to facilitate the insertion and removal of the probe during oxygen measurements. While this may have led to an underestimation of true oxygen estimates, the qualitative differences among treatments should remain robust. Furthermore, using the relative change in oxygen over a fixed period as a proxy for oxygen consumption helps minimise the influence of varying initial oxygen concentrations caused by differences in temperature among treatments (e.g., differences in dissolution rates).

Two additional control treatments were included, each crossed with the three temperatures (4 replicates per treatment): one with *Ulva* alone, and the other with both *Ulva* and *Hyale*. The former control was used to estimate the autogenic change in algal mass, while the latter assessed the natural mortality of *Hyale* and the amount of *Ulva* consumed by the amphipods. Both the mortality of *Hyale* (which was zero) and its consumption of *Ulva* (< 0.01 g) over a 24-hour period were negligible. The consumption of *Ulva* by *Palaemon* in the main experiment was corrected for autogenic change in algal mass in the absence of *Palaemon* and *Hyale*.

The experiment complies with national law for the protection of animals (DL nº 133/2013) and no ethical approval was necessary.

### 2.3. Statistical analysis

Data were analysed using a two-way analysis of variance (ANOVA), with Treatment and Temperature as fixed factors. Prior to analysis, the heterogeneity of variances was tested using Cochran's test, and transformations were applied where necessary [42]. Because two-way ANOVA comparisons of *Ulva* consumption and *Hyale* consumption shared a common treatment level where both prey species were present (i.e., the "choice" treatment was compared against *Ulva* alone for one analysis and against *Hyale* alone for the other), alpha was corrected to 0.025 to account for multiple tests. Student-Newman-Keuls (SNK) tests were conducted to assess differences among levels within significant ANOVA terms.

## 3. Results

Prawn mass ranged between 0.07 and 0.15 g (dry mass) and did not differ significantly among treatments or temperatures (treatment: $F_{2,63}=0.70$, P=0.500; temperature: $F_{2,63}=0.04$, P=0.961; interaction: $F_{4,63}=1.00$, P=0.413).

Consumption of *Ulva* by *Palaemon* was influenced by both treatment and temperature (interaction: $F_{2,42}=5.81$, P=0.0059). Inspection of SNK tests revealed that, when no choice was provided, *Palaemon* consumption of *Ulva* averaged 0.065±0.002 g, with no significant differences across temperatures (Fig 2a). However, when *Palaemon* had a choice between *Ulva* and *Hyale*, the consumption of *Ulva* was significantly lower at the lowest temperature compared to the other temperatures (15°C=0.044±0.003 g, 20°C=0.065±0.003 g, 25°C=0.065±0.003 g) (Fig 2a).

The consumption of *Hyale* by *Palaemon* was also influenced by both treatment and temperature (interaction: $F_{2,42}=3.32$, P<0.0001). SNK tests indicated that, when no choice was available, a similar number of amphipods (7.58±0.22) were consumed across all temperatures (Fig 2b). However, when given a choice, *Palaemon* consumed significantly fewer amphipods at the highest temperature compared to the other temperatures (15°C=8.25±0.25, 20°C=7.88±0.35, 25°C=5.13±0.30) (Fig 2b).

Consumption of oxygen by *Palaemon* varied significantly with temperature, independent of diet treatment (temperature: $F_{2,63}=55.68$, P<0.0001; treatment: $F_{2,63}=0.51$, P=0.6026; interaction: $F_{4,63}=1.06$, P=0.3829) (Fig 3). SNK tests

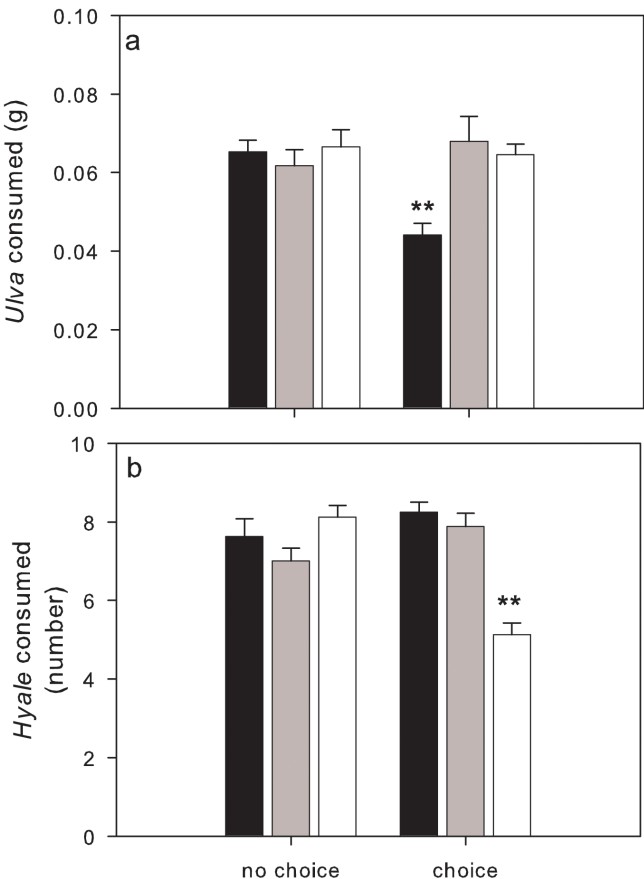

**Fig 2. Mean (+ SE, n=8) *Ulva* (a) and *Hyale* (b) consumed by *Palaemon* when either given choice or no choice across three temperatures: 15ºC black bars, 20ºC grey bars, 25ºC white bars.** ** shows a significant (P<0.01) difference as detected by SNK tests.

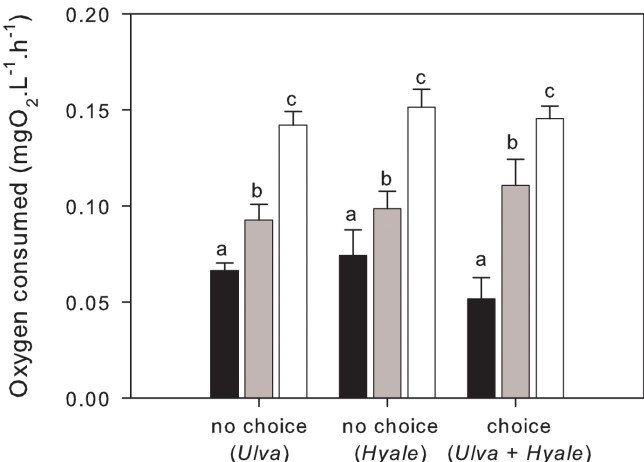

**Fig 3. Mean (+ SE, n = 8) oxygen consumption by *Palaemon* across treatments and temperatures (15°C black bars, 20°C grey bars, 25°C white bars).** Letters denote significant differences as detected by SNK tests.

showed that oxygen consumption increased significantly with rising temperature (15°C = 0.064 ± 0.006 mgO$_2$L$^{-1}$h$^{-1}$, 20°C = 0.101 ± 0.006 mgO$_2$L$^{-1}$h$^{-1}$; 25°C = 0.146 ± 0.004 mg O$_2$ L$^{-1}$ h$^{-1}$) (Fig 3).

## 4. Discussion

The results of this study demonstrate that temperature influences food preference and metabolic activity of *Palaemon elegans*, reinforcing the idea that temperature is a critical ecological factor affecting species interactions and energy flow within coastal marine ecosystems.

Our findings indicate that the consumption of *Ulva* by *Palaemon* is affected by the interaction of treatment (choice *vs.* no-choice) and temperature. When offered a choice between *Ulva* and the amphipod *Hyale*, prawns exhibited lower consumption of *Ulva* at the lowest temperature (15°C). This suggests that temperature may constrain the digestive efficiency or feeding motivation of *Palaemon* for macroalgae at cooler temperatures, possibly due to decreased metabolic rates, or via behavioural changes, most likely motivated by a decrease in *Ulva* digestive efficiency at low temperature. Conversely, at warmer temperatures (20°C and 25°C), feeding rates on *Ulva* increased in treatments where a choice was present, highlighting the potential for altered behaviour and/or enhanced digestive efficiency in *Palaemon* as temperatures rise.

The consumption of *Hyale* also varied significantly with temperature when a choice was provided, with prawns consuming fewer amphipods at the highest temperature. This behaviour may indicate a shift in feeding preferences away from animal prey and towards macroalgae at elevated temperatures. Overall, the observed results suggest that *Palaemon* may prioritize algal resources at warmer temperatures and animal resources at cooler temperatures, despite the fact that algae (*Ulva muscoides*) and animals (*Hyale perieri*) exhibit distinct nutritional profiles, with the pericarid amphipods providing a 4–5 times greater percentage of protein [43,44]. Interestingly, the transition occurred at roughly 20 °C, which is an important temperature threshold above which herbivory predominates over carnivory in other omnivorous taxa [29,33]. This suggests that temperature-driven shifts in diet may not be directly linked to the intrinsic nutritional value of the prey, but rather to a temperature-dependent ability (or inability) to digest (e.g., enzymatic digestion) certain food items.

Furthermore, the significant increase in oxygen consumption with rising temperatures underscores the metabolic challenges faced by ectothermic organisms like *Palaemon elegans*. As temperature increases, the energetic demands

of *Palaemon* also rise, likely resulting in increased foraging activity to meet these demands. This relationship aligns with existing ecological theory, which posits that temperature can modulate metabolic rates, thereby affecting trophic interactions and community structure in aquatic ecosystems [14,17,45]. In our short-term no-choice feeding experiments, however, consumption rates did not increase to match the higher metabolic rates at higher temperature. This either suggests that net energy intake could be lower in warmer conditions (i.e., the optimal temperature for net energy gain has been exceeded), or that our experimental duration was not long enough to capture relevant physiological adjustment despite the 2-wk period of acclimation. Longer-term experiments would be required to test these hypotheses and determine if such an energetic imbalance had consequences for *Palaemon* growth or reproduction.

The fact that no prawns died and all seemed to feed suggests that the experimental conditions effectively maintained the overall health of the prawns throughout the study. Moreover, the negligible effects observed in control treatments indicate that the autogenic changes in algal mass and the natural mortality of *Hyale* did not influence the experimental outcomes. This consistency allows us to attribute variations in feeding behaviour directly to temperature and food choice rather than to prey physiological stress or differences in prawn size, suggesting that the feeding responses observed in *Palaemon* were primarily driven by their behavioural choices rather than external factors affecting food availability.

This study provides valuable insights into the role of temperature in shaping the foraging behaviour of an omnivorous ectotherm, *Palaemon elegans*. As climate change continues to influence ocean temperatures, the observed patterns of feeding behaviour may have important implications for the ecological dynamics of marine communities, particularly those found in places like the Azores where sea surface temperatures are at or near 20 °C. There and elsewhere, omnivorous consumers may be near a thermal tipping point where their diets shift to include more plant and algal resources relative to animal prey, effectively lowering their trophic level and changing patterns of energy flow, interspecific competition, and top-down control. Future studies should investigate the long-term effects of sustained temperature changes on both the physiological performance and ecological interactions of *Palaemon* and other omnivorous ectotherms and their prey, particularly in the context of increasing ocean temperatures.

## Acknowledgments

We thank two anonymous reviewers for their valuable criticism and suggestions, which considerably improved the clarity of the present work.

## Author contributions

**Conceptualization:** Gustavo M. Martins, Christopher D. G. Harley, Ana Neto, Francisco Arenas.

**Data curation:** Gustavo M. Martins.

**Formal analysis:** Gustavo M. Martins.

**Funding acquisition:** Gustavo M. Martins, Christopher D. G. Harley, Ana Neto, Francisco Arenas.

**Investigation:** Gustavo M. Martins.

**Methodology:** Gustavo M. Martins, Christopher D. G. Harley, Ana Neto, Francisco Arenas.

**Resources:** Gustavo M. Martins, Ana Neto.

**Supervision:** Ana Neto.

**Validation:** Gustavo M. Martins, Christopher D. G. Harley, Francisco Arenas.

**Visualization:** Gustavo M. Martins.

**Writing – original draft:** Gustavo M. Martins.

**Writing – review & editing:** Gustavo M. Martins, Christopher D. G. Harley, Francisco Arenas.

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
