## [Decision Letter · Decision Letter 0]

15 Jul 2025

PONE-D-25-24716
Temperature-mediated shifts in feeding behaviour and metabolism in an omnivorous rock pool prawn
PLOS ONE

Dear Dr. Harley,

Thank you for submitting your manuscript to PLOS ONE. After careful consideration, we feel that it has merit but does not fully meet PLOS ONE’s publication criteria as it currently stands. Therefore, we invite you to submit a revised version of the manuscript that addresses the points raised during the review process.

We look forward to receiving your revised manuscript.

Kind regards,

Mehrnoush Aminisarteshnizi, Ph.D.

Academic Editor

PLOS ONE

Journal Requirements:

“Funding was provided from National Funds through FCT (Fundação para a Ciência e a Tecnologia) under the projects UID/BIA/00329/2013 and UIDB/50027/2020. GMM was supported by a post-doctoral grant (SFRH/BDP/108114/2015) awarded by FCT. FA received additional funding from the project SEEINGSHORE (NORTE-01-0145-FEDER-031893), co-financed by NORTE 2020, Portugal 2020 and the European Union through the ERDF, and by FCT through national funds. CDGH was supported by a Natural Sciences and Engineering Research Council (Canada) Discovery Grant (RGPIN-2022-04683).”

Additional Editor Comments:

Dear Author,

Thank you for submitting your manuscript entitled “Temperature-mediated shifts in feeding behaviour and metabolism in an omnivorous rock pool prawn” to PLOS ONE.

After careful consideration of the reviewers' comments and your responses, I have decided that your manuscript requires minor revision before it can be accepted for publication. The reviewers found your study interesting and valuable, but they have suggested a few minor points that need to be addressed.

Please revise your manuscript accordingly and provide a detailed response to each reviewer comment. Once the revisions are received, I will review them and proceed with the next steps.

Thank you for your contribution, and I look forward to receiving your revised manuscript.

Best regards,

Mehrnoush Aminisarteshnizi, Ph.D.

Academic Editor

Reviewers' comments:

Reviewer's Responses to Questions

**Comments to the Author**

1. Is the manuscript technically sound, and do the data support the conclusions?

Reviewer #1: Partly

Reviewer #2: Yes

2. Has the statistical analysis been performed appropriately and rigorously? 

Reviewer #1: Yes

Reviewer #2: Yes

3. Have the authors made all data underlying the findings in their manuscript fully available?

Reviewer #1: Yes

Reviewer #2: Yes

4. Is the manuscript presented in an intelligible fashion and written in standard English?

Reviewer #1: Yes

Reviewer #2: Yes

5. Review Comments to the Author

Reviewer #1: PLOS ONE

PONE-D-25-24716

Introduction:

Most references used are old. There is vast recent research that has been done on the relationship between temperature and feeding mechanisms.

Add a paragraph on the introduction section highlighting an overview of the feeding habits of Palaemon elegans feeding habits in the wild/nature. This will help us understand the feed preference in the experimental setup

Elaborate how the samples (Palaemon elegans) were collected and transported (Line 116-117)_Laboratory experiment

Line 119: Highlight the “larger individual” size or the size used.

Line 123-126: Elaborate the total number of samples used

Line 148: Explain how they were “killed” or rather rephrase the word “killed”

Line 119: “Haphazardly”? Describe the experimental designed employed in the study.

Line 124: Elaborate on the tank/system used. “each prawn”?

Line 126: How was the temperature monitored during water changing every other day?

Line 219: In the discussion section, it is important to state what you mean by “treatment”

Reviewer #2: Dear Authors

I have checked the paper entitled " Temperature-mediated shifts in feeding behaviour and metabolism in an omnivorous rock pool prawn". Despite the interesting work, several concerns about this MS need to be reconsidered by the Authors.

This manuscript addresses an ecologically relevant and timely question regarding the effects of temperature on trophic behavior and metabolic rate in the omnivorous prawn Palaemon elegans. The experimental design is sound, statistical analyses are appropriate, and the results are clearly presented and discussed in light of current literature. The findings have important implications for understanding potential ecological shifts in intertidal systems under climate change scenarios.

Major Strengths

The study is well-conceived and directly tests a clear hypothesis grounded in ecological theory.

Experimental conditions realistically reflect natural temperature ranges observed in Azorean intertidal pools.

The use of both no-choice and choice feeding treatments allows meaningful inference on dietary preferences.

Results are well-supported by statistical analysis and integrated with relevant literature on ectotherm physiology and trophic dynamics.

Suggestions for Improvement

While the manuscript is overall strong, I suggest the following revisions to improve clarity, context, and interpretability of the results:

Clarify novelty and knowledge gap

The introduction should better emphasize what is novel about this study. Several similar studies have explored temperature effects on omnivorous behavior in other taxa. What does this study add specifically with respect to P. elegans, Azorean ecosystems, or omnivore plasticity? Consider framing the study more explicitly within a clear gap in knowledge.

Include nutritional/ecological context of prey

The prey items (Ulva muscoides and Hyale perieri) are ecologically relevant, but the manuscript lacks information on their nutritional composition or digestibility. Given that feeding decisions often reflect food quality, please include brief comparative information (e.g., protein, fiber, or energy content) or acknowledge this as a limitation.

Address lack of proportional increase in food intake

Oxygen consumption increased substantially with temperature, yet food consumption did not increase proportionally in the no-choice treatments. Please elaborate on this mismatch. Could it suggest an energetic imbalance, suboptimal feeding, or limitations in digestion rate at higher temperatures?

Clarify biological characteristics of prawns

The sex and reproductive status of the experimental animals are not specified. Given that sex or maturation could affect feeding behavior, please clarify whether individuals were all of the same sex/stage, or acknowledge potential variability.

Short experimental duration

Feeding trials lasted 24 hours. While sufficient for detecting short-term shifts, longer exposure might be necessary to capture physiological acclimation or changes in growth. Consider noting this as a limitation and recommending longer-term trials in future research.

Figure annotations

Please ensure that sample sizes (n) are noted within figure captions or graphs for clarity. This small addition would aid reader interpretation without detracting from the visual simplicity.

Minor Comments

Line 63: "determine" → should be "determining"

Line 86: "effeciency" → should be "efficiency"

Line 220: "altered behavioural most likely..." → unclear phrasing. Consider rewording for clarity.

Line 276: "Acknowledgemnts" → spelling error; should be "Acknowledgments"

Conclusion and Recommendation

This manuscript presents a well-executed study that offers valuable insight into how temperature modulates feeding preferences and metabolic responses in omnivorous marine consumers. With minor clarifications and editorial improvements, the paper will be suitable for publication in PLOS ONE.

6. PLOS authors have the option to publish the peer review history of their article (what does this mean?). If published, this will include your full peer review and any attached files.

Reviewer #1: No

Reviewer #2: No

---

## [Author Response · Author response to Decision Letter 1]

3 Oct 2025

[See also our response to reviewers document.]

REVIEWER #1

Introduction:

Most references used are old. There is vast recent research that has been done on the relationship between temperature and feeding mechanisms.

R: Following this reviewer, and also reviewer#2 input, we have revised some parts of the introduction. While doing so, we included more recent examples available in the literature. We hope that this reviewer finds these changes appropriate.

Add a paragraph on the introduction section highlighting an overview of the feeding habits of Palaemon elegans feeding habits in the wild/nature. This will help us understand the feed preference in the experimental setup

R: Although we briefly described the ecology, including feeding, of Palaemon in a specific section in the methods, we added the following lines to the introduction as per suggestion. “The prawn P. elegans is a widespread species in coastal waters of Europe where it feeds on a diverse array of items, including detritus, algal material (such as Cladophora, Ectocarpales and Ulva) and animals (such as gammarids, harpaticoids, chironomids and ostracods)(Janas & Baránska 2008, Persson et al. 2008).”

Elaborate how the samples (Palaemon elegans) were collected and transported (Line 116-117)_Laboratory experiment

R: We added some more detail as suggested.

“Individuals of Palaemon elegans were collected with a handnet from intertidal rockpools in Lagoa (37º 44’ N, 25º 35’ W), kept in water and promptly transported to the laboratory inside coolers.”

Line 119: Highlight the “larger individual” size or the size used.

R: We added the following “There, they were sorted by size, with small individuals (< 5 mm carapace length) being discarded. Larger individuals (those with a carapace length > 5 mm) were retained and haphazardly divided into three groups to ensure similar prawn sizes in each group”

Line 123-126: Elaborate the total number of samples used

R: Not sure what the reviewer means here. The total number of replicates used is already described below where it reads “The experiment had three treatments where Ulva (no choice), Hylae (no choice) or a mixture of Ulva and Hyale (with choice) were offered to prawns, with each treatment crossed with three temperature conditions (15, 20 and 25 ºC). There were 8 replicates per treatment combination, resulting in a total of 72 experimental units.” That is, we used a total of 72 prawns.

Does the reviewer refer to the total number of prawns collected including those that were used in the experiment (72) and those that were left unused (replacements and moulted)? If so, we did not track these exact numbers, but they were around 100 (total number of prawns kept). This information was included in the text.

Line 148: Explain how they were “killed” or rather rephrase the word “killed”

R: We added the following “After the experiment, the prawns were euthanised by lowering temperature to 4ºC until insensible and then rapidly cutting through the centreline of the head and tail.”

Line 119: “Haphazardly”? Describe the experimental designed employed in the study.

R: We revised the sentence following this suggestion which now reads “Larger individuals (those with a carapace length > 5 mm) were retained (n = 100 individuals) and haphazardly divided into three groups, visually ensuring that each group had a similar range of sizes”

Line 124: Elaborate on the tank/system used. “each prawn”?

R: Not exactly sure that other information is requested here that is not already described in the text. We added “each individual prawn was kept in an individual 0.5L tank filled with continuously aerated seawater and fed ad libitum with commercial shrimp food (Hikari). The tanks were cleaned, and water was replaced every other day.

Line 126: How was the temperature monitored during water changing every other day?

R: we added the following information “The tanks were cleaned, and water (previously conditioned to the corresponding treatment’s temperature) was replaced every other day.”

Line 219: In the discussion section, it is important to state what you mean by “treatment”

R: We added the following “Our findings indicate that the consumption of Ulva by Palaemon is affected by the interaction of treatment (choice vs. no-choice) and temperature”

REVIEWER #2

Reviewer #2: Dear Authors

I have checked the paper entitled " Temperature-mediated shifts in feeding behaviour and metabolism in an omnivorous rock pool prawn". Despite the interesting work, several concerns about this MS need to be reconsidered by the Authors.

This manuscript addresses an ecologically relevant and timely question regarding the effects of temperature on trophic behavior and metabolic rate in the omnivorous prawn Palaemon elegans. The experimental design is sound, statistical analyses are appropriate, and the results are clearly presented and discussed in light of current literature. The findings have important implications for understanding potential ecological shifts in intertidal systems under climate change scenarios.

Major Strengths

The study is well-conceived and directly tests a clear hypothesis grounded in ecological theory.

Experimental conditions realistically reflect natural temperature ranges observed in Azorean intertidal pools.

The use of both no-choice and choice feeding treatments allows meaningful inference on dietary preferences.

Results are well-supported by statistical analysis and integrated with relevant literature on ectotherm physiology and trophic dynamics.

Suggestions for Improvement

While the manuscript is overall strong, I suggest the following revisions to improve clarity, context, and interpretability of the results:

Clarify novelty and knowledge gap

The introduction should better emphasize what is novel about this study. Several similar studies have explored temperature effects on omnivorous behavior in other taxa. What does this study add specifically with respect to P. elegans, Azorean ecosystems, or omnivore plasticity? Consider framing the study more explicitly within a clear gap in knowledge.

R: we agree that since we did this experimental work, a few other papers have also explored these ideas, although these had not been published yet when we did the work back in 2018, but which, for personal reasons, results were kept in a drawer until now.

Having said this, we added a few lines to the introduction to better set this manuscript in context to the wider literature available “Over the past years, a few studies have experimentally explored this idea that temperature can affect feeding behaviour, and while ectothermic herbivores and carnivores generally ingest more food with rising temperature (e.g., see review by Brown et al. 2004), omnivorous ectotherms may respond in two different ways: quantitatively by consuming more food and qualitatively by altering their degree of herbivory or carnivory through diet shifts. Shifts in diet in omnivorous ectotherms have been implied by seasonal variations in trophic level (using stable isotope analysis)(e.g., Boersma et al. 2016), by recording changes in the amount of different prey items ingested in separate with rising temperature (e.g., Behrens & Lafferty 2007, Zhang et al. 2020), or by using choice vs. no-choice experiments, but usually using dead animal prey (Zhang et al. 2018, Carreira et al. 2016), which may be unrealistic in a natural setting.

In this study, we investigated the effect of temperature on respiration rates and algal vs. animal prey preferences of the omnivorous rock pool prawn Palaemon elegans Rathke, 1837, using live prey.”

Include nutritional/ecological context of prey

The prey items (Ulva muscoides and Hyale perieri) are ecologically relevant, but the manuscript lacks information on their nutritional composition or digestibility. Given that feeding decisions often reflect food quality, please include brief comparative information (e.g., protein, fiber, or energy content) or acknowledge this as a limitation.

R: We found one study detailing the nutrient profile for Ulva (Enteromorpha) muscoides in the Canary Islands (similar setting to that of the Azores), and one study doing the nutuirent profile of several amphipod species, including Hyale perieri, in Gilbraltar and which showed that all amphipods had a similar nutrient profile. Using these studies, it is possible to compare the % content of proten and carbohidrates which we included in the text as follows: “Overall, the observed results suggest that Palaemon may prioritize algal resources at warmer temperatures and animal resources at cooler temperatures. Despite the fact that algae (Ulva muscoides) and animals (Hyale perieri) exhibit distinct nutrient profiles, with the pericarid amphipods providing 4-5 times greater percentage of protein (Granado & Caballero 2001, Baeza-Rojano et al. 2014), interestingly, the transition occurred at roughly 20 °C, which is an important temperature threshold above which herbivory predominates over carnivory in other omnivorous taxa (Vejrikova et al. 2016, Zhang et al. 2020). This suggests that temperature-driven shifts in diet may not be directly linked to the nutritional value of the prey, but rather to a temperature-dependent ability (or inability) to digest (e.g. enzymatic digestion) certain food items.”.

Address lack of proportional increase in food intake

Oxygen consumption increased substantially with temperature, yet food consumption did not increase proportionally in the no-choice treatments. Please elaborate on this mismatch. Could it suggest an energetic imbalance, suboptimal feeding, or limitations in digestion rate at higher temperatures?

R: It is true that there was an imbalance, which we already pointed out in the our discussion as follows (from original submission “Furthermore, the significant increase in oxygen consumption with rising temperatures underscores the metabolic challenges faced by ectothermic organisms like Palaemon elegans. As temperature increases, the energetic demands of Palaemon also rise, likely resulting in increased foraging activity to meet these demands. This relationship aligns with existing ecological theory, which posits that temperature can modulate metabolic rates, thereby affecting trophic interactions and community structure in aquatic ecosystems (Savage et al. 2004, Voigt et al. 2003, Zheng et al. 2021). In our short-term no-choice feeding experiments, however, consumption rates did not increase to match the higher metabolic rates at higher temperature. This either suggests that net energy intake could be lower in warmer conditions (i.e., the optimal temperature for net energy gain has been exceeded), or that our experimental duration was not long enough to capture relevant physiological adjustment despite the 2-wk period of acclimation. Longer-term experiments would be required to test these hypotheses and determine if such an energetic imbalance had consequences for Palaemon growth or reproduction.” We feel that we cannot really say much more about this other than we already discussed without speculating.

Clarify biological characteristics of prawns

The sex and reproductive status of the experimental animals are not specified. Given that sex or maturation could affect feeding behavior, please clarify whether individuals were all of the same sex/stage, or acknowledge potential variability.

R: We had no specific hypothesis regarding how sex of prawns should affect shifts in feeding in relation to temperature so we did not consider the sex of the prawns. We did, however, deliberately discarded the smallest animals (< 5 mm carapace legnth) to ensure that only adult individuals were used. Having said this, any potential effect of sex in the prawns’ diet is randomly distributed among treatments so that it likely wont change the results. This was acknowldeged in the text as follows “Since there was no hypothesis regarding how temperature-dependent changes in diet should vary with the sex of prawns, these were sorted among treatments disregarding their sex. Any potential effect of sex was thus randomly distributed among treatments, so that it was unlikely a confounding factor.”

Short experimental duration

Feeding trials lasted 24 hours. While sufficient for detecting short-term shifts, longer exposure might be necessary to capture physiological acclimation or changes in growth. Consider noting this as a limitation and recommending longer-term trials in future research.

R: it is already recommended. Please see lines 250-251 in the original submission, which were kept as is in the revised version.

Figure annotations

Please ensure that sample sizes (n) are noted within figure captions or graphs for clarity. This small addition would aid reader interpretation without detracting from the visual simplicity.

R: The number of replicates (n = 8) was included in the figure captions as suggested.

Minor Comments

Line 63: "determine" → should be "determining"

R: revised as suggested

Line 86: "effeciency" → should be "efficiency"

R: sentence revised

Line 220: "altered behavioural most likely..." → unclear phrasing. Consider rewording for clarity.

R: revised as suggested

Line 276: "Acknowledgemnts" → spelling error; should be "Acknowledgments"

R: revised

Conclusion and Recommendation

This manuscript presents a well-executed study that offers valuable insight into how temperature modulates feeding preferences and metabolic responses in omnivorous marine consumers. With minor clarifications and editorial improvements, the paper will be suitable for publication in PLOS ONE.

R: Thank you

---

## [Editor Report · Decision Letter 1]

19 Oct 2025

Temperature-mediated shifts in feeding behaviour and metabolism in an omnivorous rock pool prawn

PONE-D-25-24716R1

Dear Dr. Harley,

We’re pleased to inform you that your manuscript has been judged scientifically suitable for publication and will be formally accepted for publication once it meets all outstanding technical requirements.

Kind regards,

Mehrnoush Aminisarteshnizi, Ph.D.

Academic Editor

PLOS ONE

Additional Editor Comments (optional):

Both reviewers recommended minor revisions, and the authors have provided satisfactory responses and revisions. The paper meets the journal’s standards for scientific rigor and clarity. I recommend acceptance. I have no conflict of interest.

---

## [Editor Report · Acceptance letter]

PONE-D-25-24716R1

PLOS ONE

Dear Dr. Harley,

I'm pleased to inform you that your manuscript has been deemed suitable for publication in PLOS ONE. Congratulations! Your manuscript is now being handed over to our production team.

Kind regards,

on behalf of

Dr. Mehrnoush Aminisarteshnizi

Academic Editor

PLOS ONE